# Study on Deformation Characteristics of Retaining Structures under Coupled Effects of Deep Excavation and Groundwater Lowering in the Affected Area of Fault Zones

**Yungang Niu [1], Liang Zou [2], Qiongyi Wang [3] and Fenghai Ma [1,3,*]**

1   College of Architecture and Engineering, Dalian University, Dalian 116622, China; nyg@dlu.edu.cn
2   Shenzhen Dasheng Surveying Technology Co., Shenzhen 518000, China
3   School of Mechanics and Engineering, Liaoning Technical University, Fuxin 123000, China
*   Correspondence: xkb@vip.163.com

**Abstract:** In order to study the deformation characteristics of the retaining structure under the coupled effect of excavation and dewatering in the affected area of fault zones, this paper takes a deep excavation project in the F1322 fault zone influence area in Shenzhen as an example. The research methods of theoretical analysis, numerical simulation and field measurement are used to conduct in-depth research on the deformation of the retaining structure caused by the excavation and dewatering of the foundation pit. The results show that considering the coupled effect of dewatering in the foundation pit, the energy method based on elastic theory is more accurate in solving the deformation of the retaining pile. By comparing and analyzing the theoretical calculation results, numerical analysis results, and field measurement values, we found that the numerical laws of the three are basically the same. Simplified calculations that only consider rotational deformation and ignore the translational deformation of the wall lead to large deviations between the theoretical calculation results and the measured values of the wall bottom deformation. In order to reduce the deviation between numerical results and measured values, the construction of the foundation pit should strictly adopt measures such as "sectional excavation, avoiding peripheral loads, and optimizing construction deployment", strengthen construction monitoring, and reduce the impact on the deformation of the retaining pile. The maximum deformation growth rate $k$ ($\Delta S_{max}/\Delta$) of the retaining pile decreases approximately exponentially with the increase of the structural stiffness parameters ($E$ and $I$) and the embedment ratio within a certain range. The sensitivity analysis of the lateral displacement of the retaining pile to different geological parameters is conducted, and the sensitivity factors of the geological parameters to the deformation of the retaining structure are obtained, namely the maximum internal friction angle, followed by the cohesion, and the elastic modulus is the smallest. Based on the original design plan, an optimization of the excavation design is proposed by reducing the stiffness of the support structure. Therefore, the research findings in this paper have significant theoretical and practical implications for the engineering design of excavation projects located in fault zones. By optimizing the excavation support system, not only can standardized construction procedures be achieved, but also investment costs can be reduced, and construction time shortened, which fully aligns with the current safety, economic, and sustainable design principles of excavation projects aiming to conserve resources.

**Keywords:** fault zone influence area; coupled effect; retaining structure deformation; sensitivity analysis; optimized design

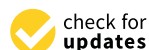



## 1. Introduction

With the rapid development of social economy and the continuous acceleration of urban construction, a large number of high-rise and super high-rise buildings have emerged, and the construction of foundation pit engineering [1–3] under complex environmental

conditions has become a hot topic in underground engineering. Fault activity not only poses serious challenges and causes significant damage to structures such as buildings and underground engineering located across fault zones [4–6], but also has a certain impact on the safety and normal use of structural engineering in the affected areas. Therefore, in-depth research on the stress and deformation characteristics of structural engineering in fault zones and their influence areas has extremely important theoretical value and engineering significance.

At present, the research results on the stress and deformation of foundation pit support structures under various geological conditions are relatively abundant both domestically and abroad [7,8], but research on the stress and deformation characteristics of foundation pit support structures in fault zone influence area is quite scarce. Therefore, it is necessary to carry out research on the stress and deformation characteristics of retaining piles under the coupled effect of foundation pit dewatering excavation in fault zone influence area. The main support structures used in deep foundation pit support technology [9,10] include gravity retaining walls, bored piles, triple tube mixed piles (SMW method piles), drilled cast-in-place piles, and underground continuous walls. Yang et al. [11] established a coupling model of foundation pit dewatering excavation and conducted numerical calculations combined with on-site measurements to analyze the deformation characteristics of the retaining piles in a deep foundation pit of a subway station in Lanzhou and concluded that simultaneous dewatering inside and outside the foundation pit was more beneficial for solving the problem of water seepage in the red sandstone. Ni et al. [12] used grey relational analysis to analyze the ultra-large deformation inducing factors of typical soft soil in Nanjing and established a weakening model based on field measurements for verification analysis. Finally, the ABC-BP neural network method was used to test the results of deformation control evaluation and prediction. Shao et al. [13] studied the prediction and analysis of the deformation of zigzag-shaped support in deep foundation pits based on neural networks, providing theoretical and analytical tools for the design of foundation pit structures. Li et al. [14] analyzed the stress and deformation characteristics of underground continuous walls, proposed a calculation formula for lateral deformation using the elastic thin plate theory method, and verified its correctness with examples. Zeng et al. [15,16] conducted indoor model experiments, focusing on the deformation characteristics of foundation pits caused by dewatering before excavation. The experiments showed that as the groundwater in the pit was pumped out, the funnel-shaped water seepage outside the pit continued to expand, causing deformation of the support structure and surface subsidence. To explore the effectiveness of internal partition walls in controlling foundation pit deformation and groundwater lowering before excavation, numerical simulations and model tests using a foundation pit pumping model were conducted and validated. Additionally, the study investigated the impact of different internal partition wall spacing on the control of foundation pit deformation. Hu et al. [17] conducted single-factor risk evaluation and hierarchical fuzzy comprehensive evaluation of actual engineering, and proposed recommendations for controlling foundation pit deformation. Chen et al. [18] obtained on-site data of lateral and vertical displacement of retaining piles and peripheral ground subsidence by monitoring the excavation process of a foundation pit group, analyzed and compared the deformation characteristics of the deep foundation pit group and the interaction between foundation pits, and proposed optimization measures for deformation control. Cao et al. [19] studied the control technology for deformation of super-deep excavation and investigated deformation control measures such as pre-stressing technology and post-strengthening technology. Yu [20] used finite element analysis to design and optimize the form of excavation support systems such as straight beam support system and large open-loop beam support system, providing valuable references for excavation support design. Tian et al. [21] compared and selected the optimal support design scheme for excavation support in five main aspects. An et al. [22] used the fuzzy logic method to conduct an economic analysis and evaluation of the subway station support structure to determine the optimal solution. Savvides and Papadopoulos formed a Feed Forward

Neural Network that estimate failure stresses and strains in Shallow Foundations formed through Stochastic Finite Element Analysis following Savvides and Papadrakakis [23,24].

Based on the analysis of the current research status of scholars on excavation engineering in the literature mentioned above, it can be seen that there is almost no research on the deformation characteristics of excavation support piles in deep excavation projects under the coupling effect of excavation and dewatering in the influence area of fault zones. This paper takes the deep excavation project in Luohu District, Shenzhen, which is located in the impact area of the fault zone, as the research background. The deformation characteristics of excavation support piles under the coupling effect of excavation and dewatering in deep excavations are studied using theoretical calculations, numerical simulations, and on-site measurements. A theoretical analytical solution for the lateral deformation of the support structure considering the coupling effect of excavation dewatering is proposed, and the influence of structural design parameters and various geological parameters on its deformation is analyzed. Based on the original plan, an optimized design direction is proposed.

## 2. Background

### 2.1. Project Overview

This project is located southeast of the Yijing Station on Line 5 of the Shenzhen Metro in Luohu District, Shenzhen. It is influenced by the F1322 fault zone. The fault zone is the main fault in the fault group, with a length of 38 km and a width of 7–70 m, with a maximum width of approximately 267 m. The soil parameters are selected based on the survey report, as shown in Table 1. The simplified stratigraphic structure and thickness of the site from top to bottom are: uncompacted fill soil, with a thickness of 4.4 m; clay, with a thickness of 3.05 m; angular gravel, with a thickness of 0.8 m; sandy clay, with a thickness of 4.6 m; completely weathered rock, with a thickness of 6.95 m; highly weathered rock, with a thickness of 6.90 m, moderately weathered rock, not penetrated. The excavation support safety level is level one, with a total area of approximately 8900 m², a circumference of approximately 380 m and a maximum excavation depth of 16.8 m. The support structure is composed of piles and internal supports, with two internal supports set at an interval of 8.5 m. The pile diameter of the retaining piles and the column piles is 1.2 m, the pile length of the retaining piles is 25.8 m and the pile length of the column piles is 20.0 m. The section sizes of the crown beam, waist beam, and support beam are all 1.0 m × 1.2 m. The concrete strength level of the retaining piles, support beams, connecting beams, and column piles is C30. The cross-sectional dimensions and physical-mechanical property indicators of the support structure are shown in Table 2. The plan position of the excavation engineering and the excavation cross-section diagram are shown in Figures 1 and 2.

**Table 1.** Calculation parameters of soil layer.

| Stratigraphic (Genetic) | Natural Weight (kN/m³) | Tri-Axial Test Secant Modulus/MPa | Secant Modulus of Elasticity/MPa | Unloading Modulus of Elasticity/MPa | Poisson's Ratio | Cohesion (kPa) | Internal Friction Angle (°) | Permeability Coefficient (m/d) |
|---|---|---|---|---|---|---|---|---|
| ① Uncompacted fill soil | 19.0 | 2 | 2 | 6 | 0.35 | 14 | 15 | 5 |
| ② Clay | 18.5 | 2.8 | 2.8 | 8.4 | 0.33 | 25 | 18 | 0.1 |
| ③ Angular gravel | 19.5 | 6 | 6 | 18 | 0.28 | 0 | 36 | 25 |
| ④ Sandy clay | 21.0 | 8 | 8 | 24 | 0.26 | 25 | 22 | 5 |
| ⑤ Completely weathered rock | 20.0 | 16 | 16 | 48 | 0.24 | 23 | 28 | 0.1 |
| ⑥ Highly weathered rock | 20.5 | 25 | 25 | 75 | 0.23 | 20 | 32 | 0.5 |

**Table 2.** Section dimensions and physical-mechanical properties of the support structure.

| Name | Material | Section Dimensions/mm | Unit Weight /(kg/m³) | Elastic Modulus/GPa | Internal Friction Angle/(°) | Poisson's Ratio |
|---|---|---|---|---|---|---|
| Retaining pile (Diaphragm wall) | C30 | Thickness 880 | 2400 | 30 | 26 | 0.2 |
| Column pile | C30 | Ø1200 | 2400 | 30 | 26 | 0.2 |
| Crown beam | C30 | 1000 × 1200 | 2400 | 30 | 26 | 0.2 |
| Support beam | C30 | 1000 × 1200 | 2400 | 30 | 26 | 0.2 |
| Waist beam | C30 | 1000 × 1200 | 2400 | 30 | 26 | 0.2 |

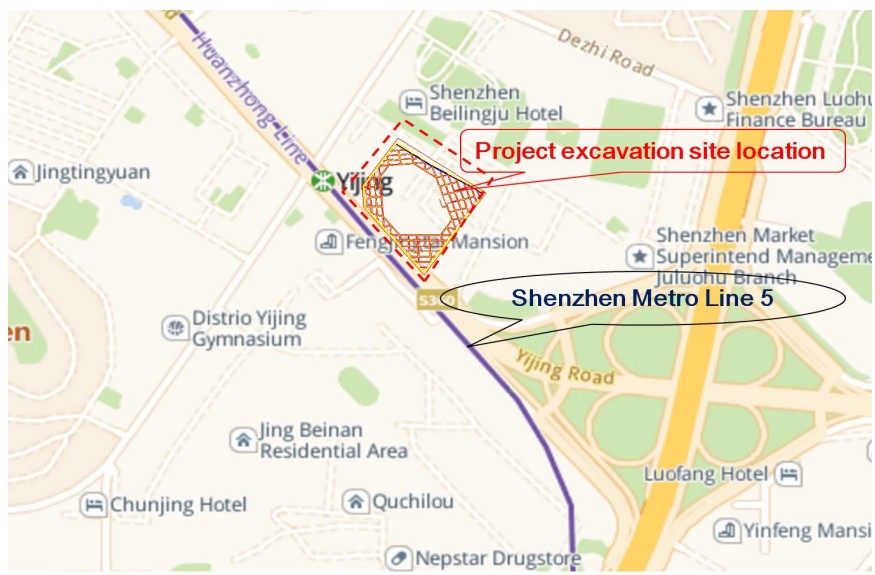

**Figure 1.** Project excavation site plane location.

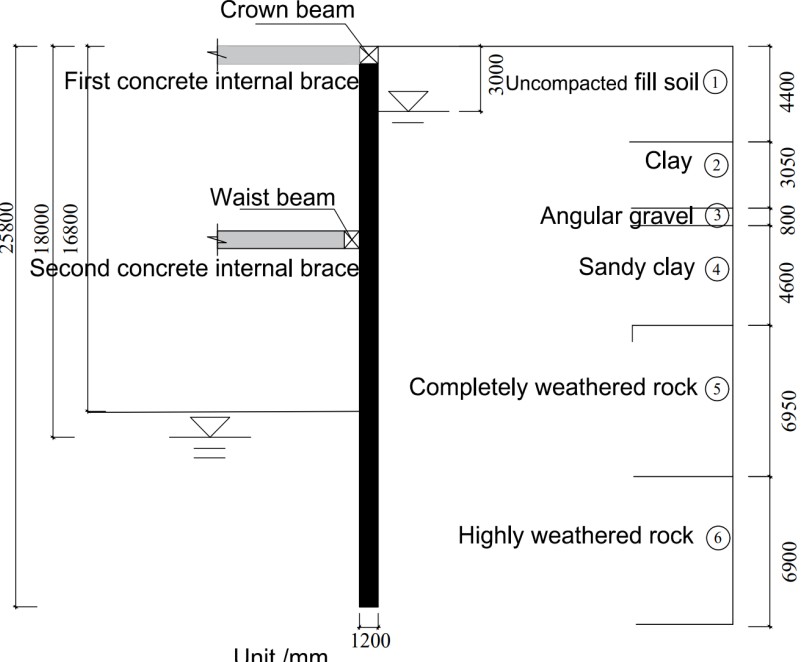

**Figure 2.** Cross-section of foundation pit excavation.

*2.2. Geological Condition*

The proposed site is located in the mixed rock of the Bijia Mountains in the Nanhua Formation. The site is affected by the F1322 (Shehecheng–Henggang–Luohu) fault (referred to as the Henggang fault). The bedrock at the site is relatively fragmented, and local green clay alteration phenomena can be seen. The stability of the area where the site is located is relatively good and belongs to a stable area. According to the analysis of the geological conditions of the fault zone and the topography and geomorphology of the site and its surroundings, the groundwater at the site consists of Quaternary porous water and bedrock fissure water. The conglomerate and medium sand layers are strong permeable layers, which are replenished laterally by atmospheric precipitation and surrounding groundwater. The bedrock fissure water mainly exists in the strong and moderately weathered rock joints and fissures, mainly receiving laterally infiltrating water supply from the surrounding bedrock fissure water and lateral groundwater flow from the underlying aquifer. The strong to moderately weathered rock is a moderately permeable layer.

## 3. Calculation of Deformation of Retaining Piles under the Coupled Effect of Excavation and Dewatering in Foundation Pit

In this paper, the deformation calculation of the retaining piles is based on the calculation of the equivalent underground continuous wall deformation. To simplify the calculation, the following assumptions are made: considering that the thickness of the continuous wall is much smaller than its width and height, it can be regarded as an elastic plane thin plate; ignoring the translational deformation of the bottom of the continuous wall, only considering the rotational deformation, the calculation model of the underground continuous wall can be simplified to fixed on both sides at the end of the wall, simply supported at the bottom, and free at the top; the internal support is calculated by using elastic rod elements, and the active soil pressure is calculated by using the classical Rankine soil pressure theory, and the passive soil pressure is calculated by using the linear elastic "m" method.

The calculation model of the underground continuous wall is shown in Figure 3. A rectangular coordinate system centered on the continuous wall is established, where the *x*-axis points to the direction of the wall span, the *y*-axis points downward to the height of the wall, and the *z*-axis points to the thickness direction of the excavation wall. The size of the underground continuous wall is $L \times H$, and the wall thickness is $d$. $E_a$ and $E_p$ are the active and passive soil pressures, respectively, and $E_0$ is the active soil pressure at the bottom of the excavation.

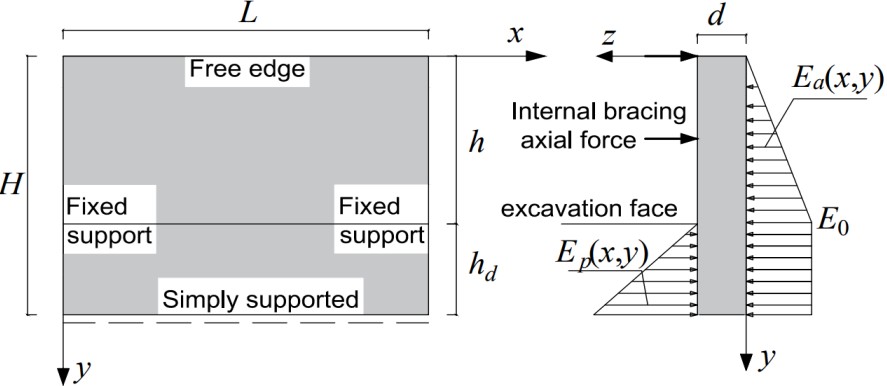

**Figure 3.** Simplified calculation model of underground diaphragm wall.

The lateral displacement of the wall is known as:

$$s = R\frac{H-y}{H}(1 - \cos\frac{2\pi x}{L})(\sin\frac{\pi x}{L} + \sin\frac{\pi y}{H}) \tag{1}$$

Based on the strain energy of the elastic plane thin plate:

$$Q = \frac{P}{2} \iint_T \left\{ \left( \frac{\partial^2 w}{\partial x^2} + \frac{\partial^2 w}{\partial y^2} \right)^2 - 2(1-\mu) \left[ \frac{\partial^2 w}{\partial x^2} \cdot \frac{\partial^2 w}{\partial y^2} - \left( \frac{\partial^2 w}{\partial x \partial y} \right)^2 \right] \right\} dxdy \tag{2}$$

where $s$ is the lateral displacement of a point on the thin plate; $R$ is the constant to be solved for; $Q$ is the strain energy of the thin plate; $P$ is the bending stiffness of the thin plate; $T$ is the integral area of the lateral wall; $E$ is the elastic modulus of the thin plate; $\mu$ is the Poisson's ratio of the thin plate.

Considering the displacement boundary conditions, the expression of the strain energy of the wall is obtained by substituting Equation (1) into Equation (2):

$$Q = PR^2 \left( \frac{388H}{L^3} + \frac{32.5L}{H^3} + \frac{60 + 41\mu}{LH} \right) \tag{3}$$

The work done by the active soil pressure, the passive soil pressure and the internal support axial force are:

$$J_1 = \iint_T E_a(x,y)s(x,y)dxdy \tag{4}$$

$$J_2 = \iint_T E_p(x,y)s(x,y)dxdy \tag{5}$$

$$J_3 = \sum_{i=1}^{n} F_i s_i \cos\theta_i = R^2 \sum_i^n \frac{K_i(H-y_i)^2}{2H^2} \cdot \left( 1 - \cos\frac{2\pi x_i}{L} \right)^2 \left( \sin\frac{\pi x_i}{H} + \sin\frac{\pi x_i}{L} \right)^2 \cos\theta_i \tag{6}$$

where $F_i$ is the axial force of the internal support $i$, $s_i$ is the deformation of the wall at the support $i$, $x_i$ and $y_i$ are the positions of the internal support, and $\theta_i$ is the angle between the normal to the excavation side and the internal support $i$; $n$ is the number of internal supports.

The potential energy is obtained as:

$$W = Q - J = Q - (J_1 - J_2 - J_3) \tag{7}$$

Substituting Equations (3)–(6) into Equation (7), the following expression is obtained:

$$W = PR^2 k_2 - RE_0 L k_1 + mLR^2 k_3 + k_4 \tag{8}$$

where the coefficient of horizontal subgrade reaction $m$ of the base soil was obtained through the back-analysis method based on field tests.

Where:

$$k_1 = \left[ \frac{Hh_d}{\pi^2 h} \sin\frac{\pi}{h} + \frac{2H^2}{\pi^3 h} + \frac{4(3Hh_d^3 + h^2)}{9\pi H} \right] L$$

$$k_2 = \frac{388H}{L^3} + \frac{32.5L}{H^3} + \frac{60 + 41\mu}{LH}$$

$$k_3 = \left( \frac{h_d^4}{12H^2} + \frac{3h_d^2}{32\pi^2} \cos\frac{2\pi h}{H} - \frac{h_d^2}{15\pi^3} \sin\frac{\pi h}{H} \right) L$$

$$k_4 = R^2 \sum_i^n \frac{K_i(H-y_i)^2}{2H^2} \cdot \left( 1 - \cos\frac{2\pi x_i}{L} \right)^2 \cdot \left( \sin\frac{\pi y_i}{H} + \sin\frac{\pi x_i}{L} \right)^2 \cos\theta_i$$

Using the Ritz method, $\partial W / \partial R = 0$, the coefficient $R$ is determined as:

$$R = \frac{k_1 E_0}{2(k_2 P + k_3 m + k_4)} \tag{9}$$

Substituting Equation (9) into Equation (8), the lateral deformation calculation expression of each point of the underground continuous wall is obtained.

$$s(x,y) = \frac{k_1 E_0}{2(k_2 P + k_3 m + k_4)} \frac{H-y}{H} \cdot (1 - \cos \frac{2\pi x}{L})(\sin \frac{\pi x}{L} + \sin \frac{\pi y}{H}) \tag{10}$$

## 4. Numerical Simulation Calculation

### 4.1. Three-Dimensional Model Establishment

The three-dimensional stress-seepage coupled analysis model was established using MIDAS GTS NX finite element software. Considering that the influence range of the excavation on the surrounding environment is about 3~5 times the excavation depth, [25] the deformation of the soil outside this range is negligible. Therefore, the numerical calculation model size is selected as: length × width × height = 225 m × 200 m × 45 m, as shown in Figure 4. It is assumed that each soil layer adopts an ideal elastoplastic model, which conforms to the modified Mohr–Coulomb strength criterion. The support structures of the foundation pit are all modeled using a linear elastic model. The foundation pit retaining piles are simulated using an equivalent earth wall model, with an equivalent thickness of 0.88 m. To better reflect the effect of the interlocking piles on seepage control, interface contact elements are used to simulate the cutoff walls in MIDAS GTS NX finite element analysis software, with the permeability coefficient set to 0, and the interface parameters generated automatically by the finite element software interface assistant. The foundation pit support system and cutoff wall arrangement are shown in Figures 5 and 6.

### 4.2. Boundary Conditions and Calculation Conditions

To accurately simulate the lateral deformation characteristics of the retaining piles under the coupled effect of excavation and dewatering of the foundation pit, horizontal constraints were applied around the model, and vertical constraints were applied at the bottom to limit horizontal and vertical displacements. The top was set as a free boundary, as shown in Figure 7. Based on the geological exploration data and specific engineering design, the actual construction conditions of the Luohu Center deep foundation pit support project were moderately simplified using MIDAS GTS NX finite element analysis software, and the analysis process of each construction stage in this simulation is shown in Table 3.

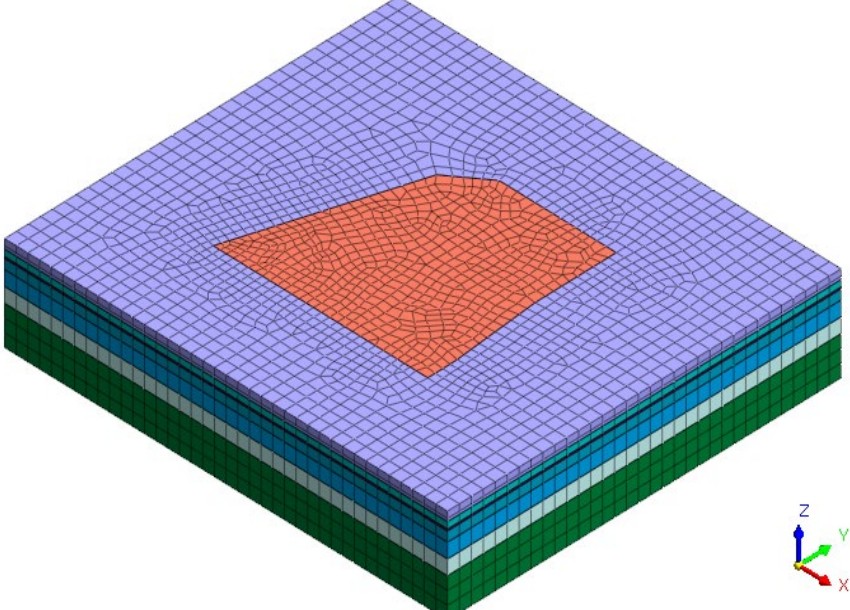

**Figure 4.** 3D numerical calculation model.

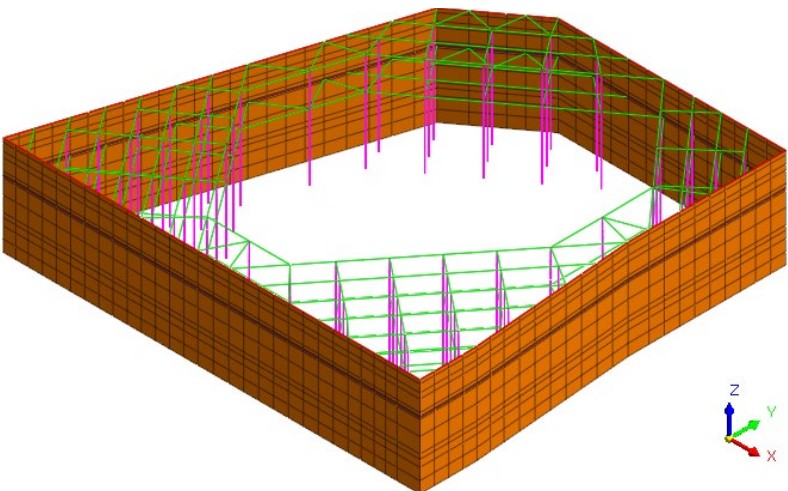

**Figure 5.** Foundation pit supporting system.

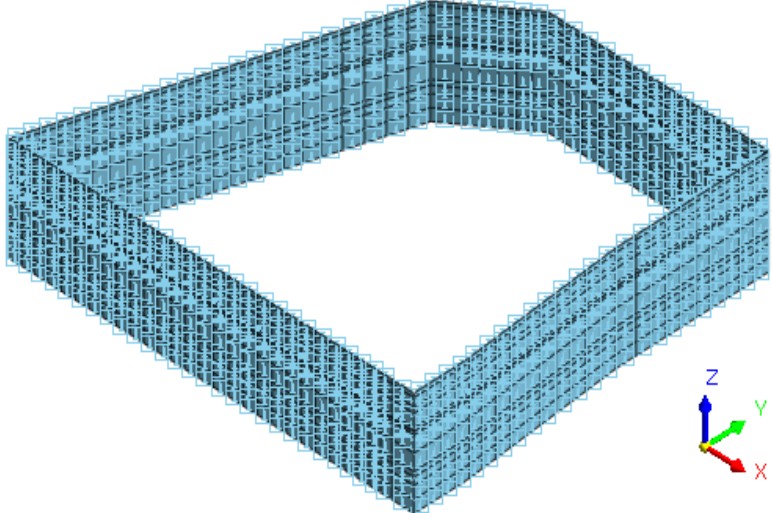

**Figure 6.** Cut-off curtain.

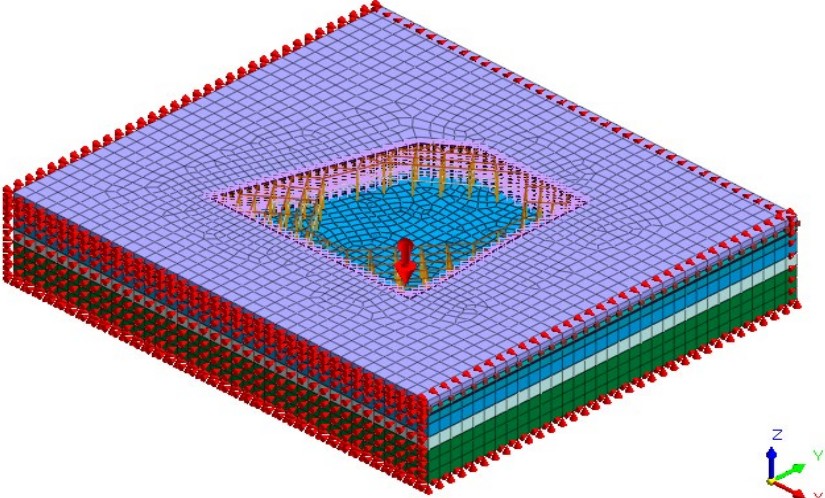

**Figure 7.** Boundary conditions and loading conditions.

**Table 3.** Construction procedure.

| Construction Stage | Construction Sequence | Working Conditions | Simulation Content |
|---|---|---|---|
| Prior to excavation of foundation pit | 1 | Initial seepage analysis | Seepage analysis before excavation of foundation pit, displacement reset. |
| | 2 | Initial stress analysis | Stress analysis before excavation of foundation pit, displacement reset |
| During excavation of foundation pit | 3 | Diaphragm wall construction | Activate foundation pit retaining piles, column piles, water cutoff curtains, and other constraints. |
| | 4 | Step 1 excavation | Dewatering the foundation pit to a depth of −9 m (initial water level −3 m). |
| | 5 | First dewatering | Excavate to a depth of −8.5 m underground, activate the second inner support, and waist beam. |
| | 6 | Step 2 excavation | Excavate to a depth of −8.5 m underground, activate the second inner support, and waist beam. |
| | 7 | Second dewatering | Dewatering to a depth of −18 m underground. |
| | 8 | Step 3 excavation | Excavate to a bottom depth of −16.8 m and activate the bottom plate. |

### 4.3. Analysis of Numerical Simulation Results

The soil pressure difference generated by the excavation and unloading of the foundation pit causes lateral displacement of the retaining piles towards the direction of the foundation pit. The curve showing the change in horizontal displacement of the retaining pile caused by excavation of the foundation pit under finite element numerical calculation is shown in Figure 8. The lateral displacement of the retaining piles has obvious regional characteristics. The lateral displacement from the top of the pile to the bottom shows a "swollen belly" curve distribution, with small lateral displacement under the constraint of internal support at the top of the pile and embedding at the bottom. The maximum deformation position is close to the excavation face. With the progress of excavation, the maximum deformation position of the pile body continuously moves downward. When the excavation is completed, the maximum deformation position of the pile body is near the lower part of the foundation pit and the maximum deformation value is 5.21 mm.

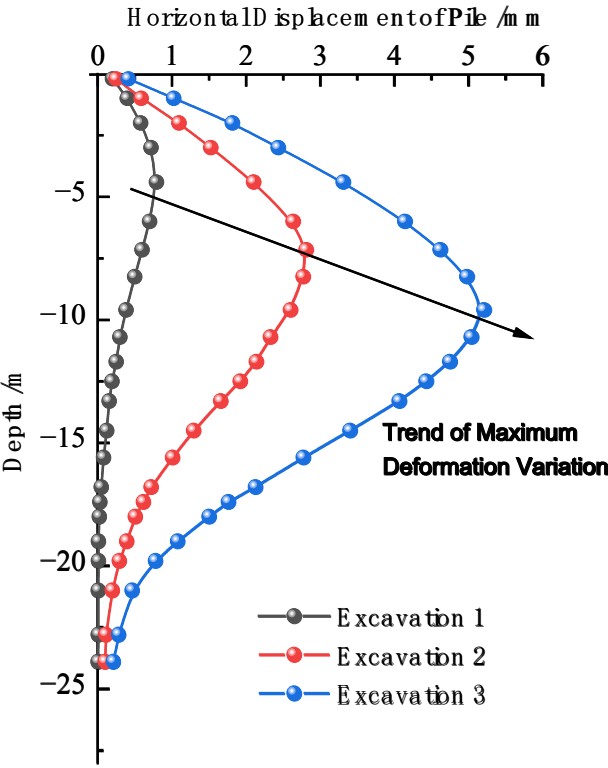

**Figure 8.** Comparison of numerical calculation results of horizontal displacement of retaining piles.

## 5. Comparison and Analysis of Numerical, Theoretical, and Field Measurement Results

The layout of the measured points in the plane of this article is shown in Figure 9. The field measurement content is the horizontal displacement of the deep retaining pile. One monitoring point is arranged for all the interlocking piles, and the results of a certain measured point on the northeast side of the excavation are selected for comparative analysis in this paper.

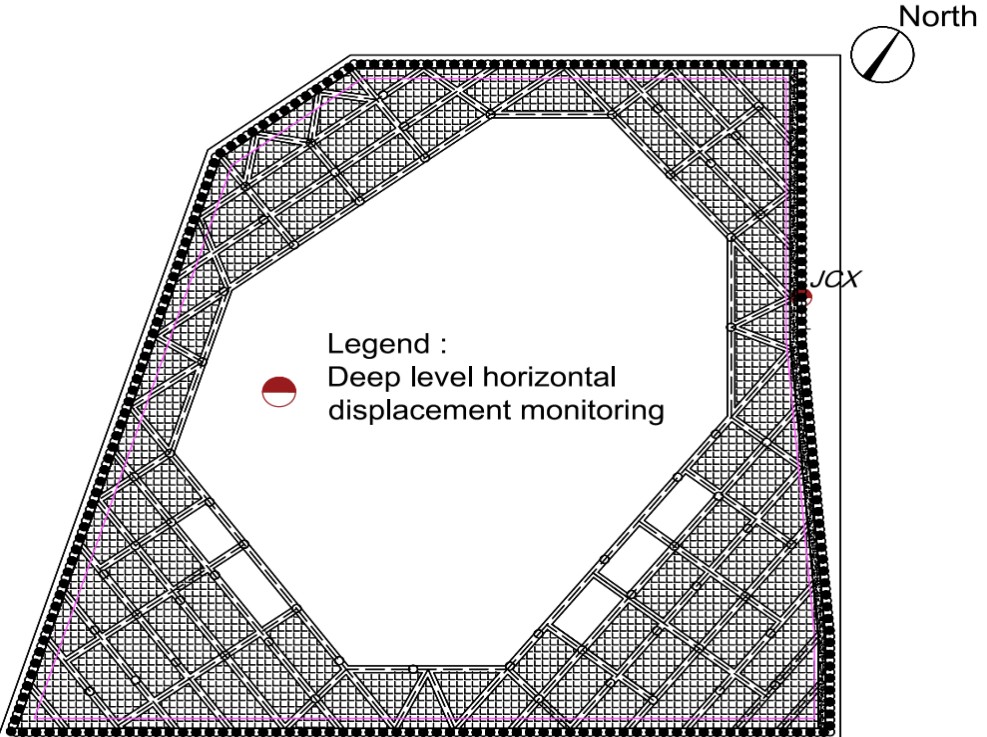

**Figure 9.** Plane layout of measured points.

The comparison and analysis of theoretical, numerical and field measurement results of the deformation of the foundation pit retaining pile are shown in Figure 10. The numerical results of the three are basically consistent, indicating that the theoretical calculation value and finite element calculation value can reflect the actual situation to a certain extent. When the excavation of the foundation pit is completed, due to the neglect of the translational deformation of the wall, and only considering the rotational deformation, the theoretical calculation result of the deformation near the bottom of the wall is significantly deviated. In fact, the elastic thin plate theory [26–28] based on small deformation theory is greatly affected by time effect [14], and when the deformation is large, the soil has undergone plastic deformation, so the theoretical calculation value is smaller than the measured value. The finite element calculation result is smaller than the measured result, which may be due to the rapid expansion of the deformation of the retaining pile caused by the excavation of the foundation pit without considering the stratification and zoning of the foundation pit, peripheral loading, and timely support reinforcement. Therefore, measures such as strict "segmentation and stratification" excavation, avoidance of peripheral loading, and optimization of construction organization should be taken to reduce the impact of foundation pit excavation on the deformation of the retaining pile.

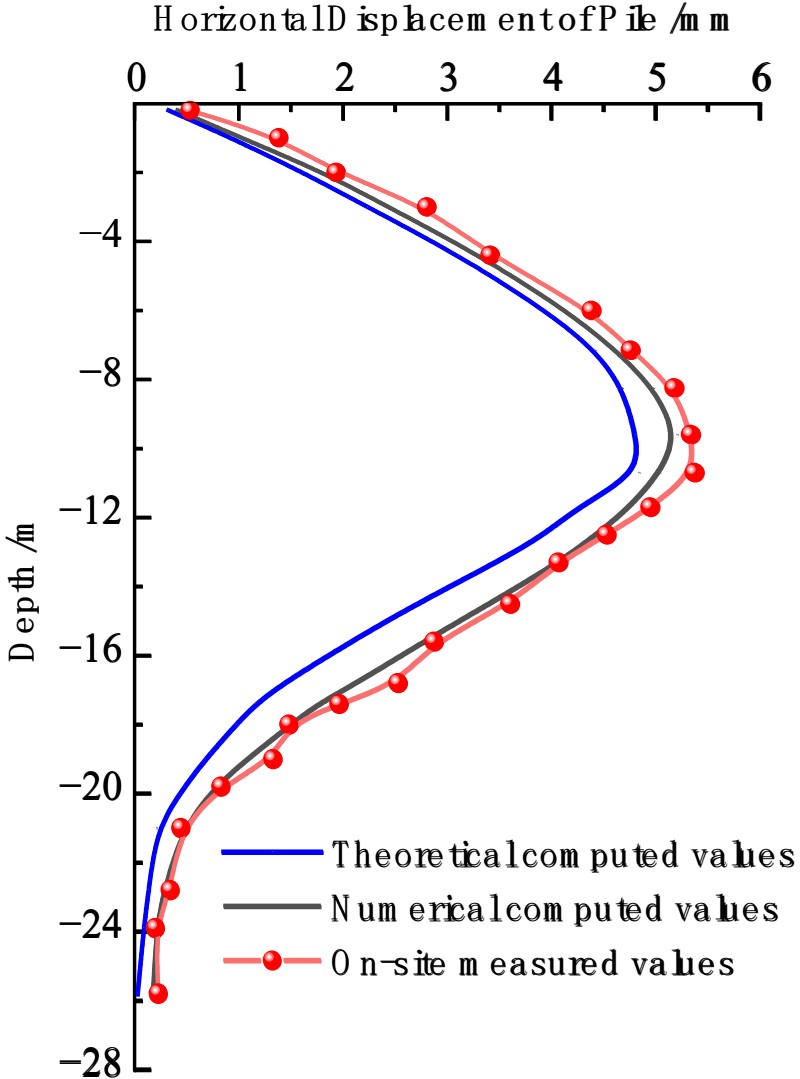

**Figure 10.** Comparison of horizontal displacement of retaining pile.

## 6. Analysis of the Influence of Structural Design Parameters and Geological Parameters

### *6.1. Analysis of the Influence of Structural Design Parameters*

6.1.1. Analysis of the Influence of Elastic Modulus

The stiffness of the retaining piles in the foundation pit is mainly influenced by the structural elastic modulus and the sectional moment of inertia [29]. By adjusting the stiffness of the retaining piles, the influence of the stiffness variation on the deformation of the retaining piles can be analyzed. The structural elastic modulus is reflected by the structural material, namely the concrete strength of the retaining piles. While keeping the other engineering design parameters constant, the concrete strength grades are sequentially adjusted to C20, C25, C30, C35 and C40, and the influence of different elastic moduli on the deformation of the retaining piles is studied. As shown in Figure 11, the lateral displacement curve of the retaining piles under different concrete strengths is obtained. With the increase of concrete strength, the elastic modulus of the retaining piles also increases, and the deformation of the retaining piles contracts. The maximum deformation position remains basically unchanged.

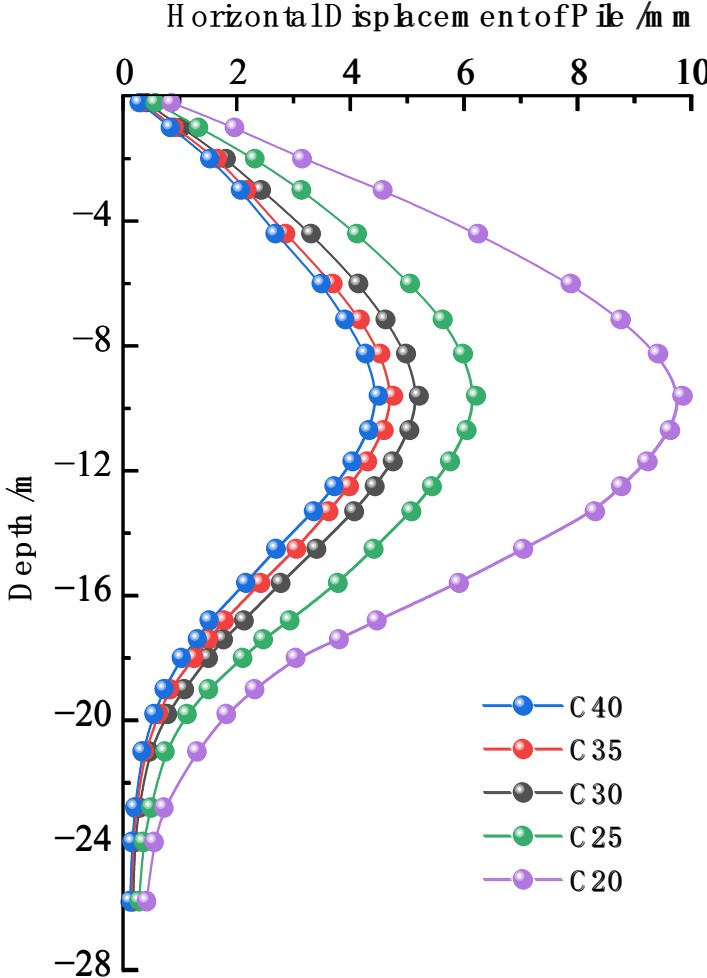

**Figure 11.** Horizontal displacement of retaining pile with different concrete strength.

To further quantify, the maximum lateral displacement value of the retaining piles under different concrete strength grades is extracted, and the influence curve of different concrete strengths on the maximum deformation of the retaining piles is obtained as shown in Figure 12. As the elastic modulus increases, the maximum deformation of the retaining piles decreases nonlinearly, and the decreasing trend is almost close to an exponential function within a certain range of elastic modulus. Meanwhile, taking E = 30 GPa of C30 as the reference, the growth rate $k$ ($\Delta S_{max}/\Delta$) of the retaining piles is introduced. The growth rate of the maximum lateral displacement of the retaining piles decreases approximately logarithmically with the increase of elastic modulus. When the concrete strength grade is low, the maximum deformation of the retaining piles will be large and significantly affected by the elastic modulus. When the concrete grade is C20, the elastic modulus is 26 GPa, and the maximum lateral displacement value of the retaining piles is 9.85 mm, showing an obvious logarithmic increase. Although it does not exceed the deformation limit value, the maximum deformation value of the retaining piles increases exponentially. When the actual construction quality is poor, the deformation value of the retaining piles will change significantly, and even lead to excessive damage, posing great hidden dangers to the stability and safety of the foundation pit support. Therefore, from this perspective, to ensure that the retaining piles have sufficient stiffness, the concrete strength of the retaining piles in this project should not be lower than C25.

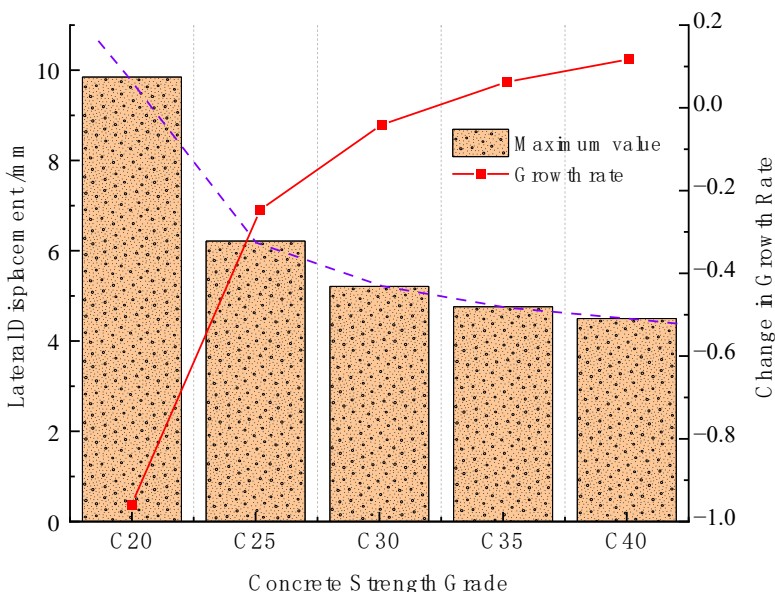

**Figure 12.** Analysis of maximum deformation of retaining pile with different concrete strength.

### 6.1.2. Analysis of the Influence of Pile Diameter

Keeping the rest of the design parameters unchanged, adjust the diameter of the retaining piles in sequence to 0.8 m, 1.0 m, 1.2 m and 1.4 m, and study the deformation law of the retaining piles under different diameters. As shown in Figure 13, obtain the lateral displacement change curve of the retaining piles under different diameters. As the diameter of the retaining piles increases, the moment of inertia also increases and the lateral displacement of the retaining piles decreases, while the position of the maximum lateral displacement remains basically unchanged.

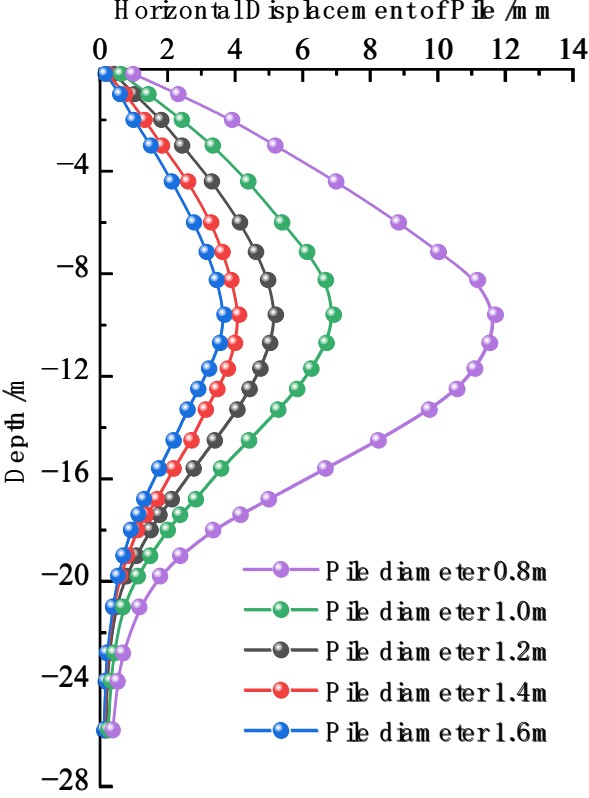

**Figure 13.** Horizontal displacement of retaining pile with different pile diameter.

To further quantify, extract the maximum lateral displacement value of the retaining piles under different diameters, and obtain the influence curve of different diameters on the maximum deformation of the retaining piles as shown in Figure 14. As the diameter increases, the maximum deformation of the retaining piles decreases approximately in a nonlinear function, and within a certain range of diameters, the curve is approximately an exponential function. Taking the section inertia distance I under a pile diameter of 1.0 m as the baseline, the growth rate $k$ ($\Delta S_{max}/\Delta$) of the maximum lateral displacement of the retaining piles is introduced. The growth rate decreases approximately in a logarithmic function with the increase of the diameter. When the diameter is 0.8 m, the maximum lateral displacement of the retaining piles is 7.62 mm. When the diameter is further reduced, the increasing trend of the maximum lateral displacement approximately follows a logarithmic function, and the deformation of the retaining piles increases sharply, exceeding the deformation control range, and resulting in engineering quality and safety problems caused by the too-small stiffness of the retaining piles. From this point of view, it is considered that the diameter of the retaining pile affects its stiffness, and the diameter should not be less than the limit of 0.8 m to ensure the safety stiffness reserve of the retaining pile.

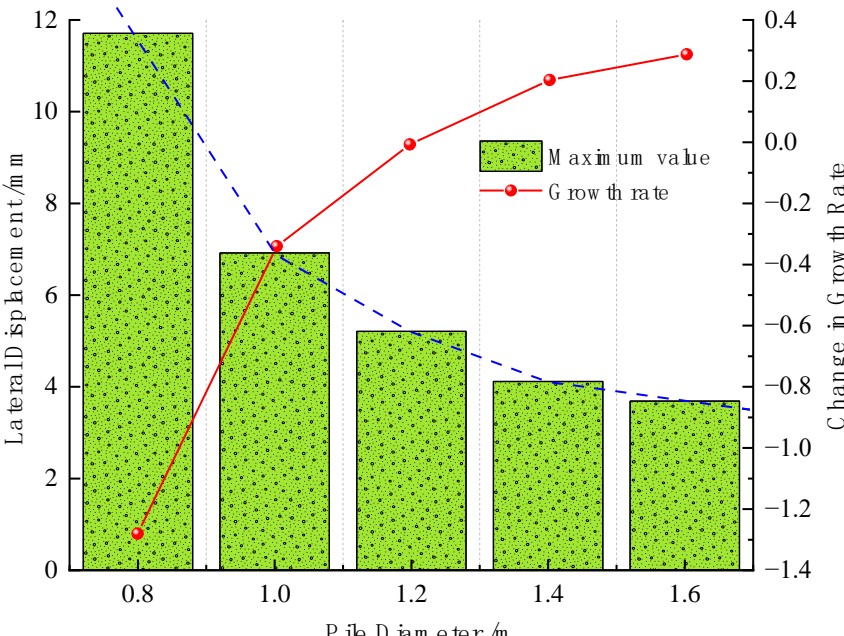

**Figure 14.** Analysis of maximum deformation of retaining pile with different pile diameter.

6.1.3. Analysis of the Influence of Embedment Ratio

Keeping the other engineering design parameters unchanged, adjust the embedment ratio(depth of pile embedment/depth of excavation of foundation pit) to 0.18, 0.30, 0.42, 0.53 and 0.65 in sequence, and study the deformation law of the retaining piles under different embedment ratios. As shown in Figure 15, obtain the lateral displacement change curve of the retaining piles under different embedment ratios. As the embedment ratio increases, the lateral displacement of the retaining piles decreases continuously, and the position of the maximum lateral displacement gradually moves up. When the embedment depth reaches a certain depth (embedment ratio of 0.65), under the condition of constant stiffness of the retaining piles, the lateral displacement changes very little or remains unchanged. When the embedment ratio approaches 0.42, there is a significant increase in the lateral displacement of the retaining pile. When the embedment depth of the retaining piles is further reduced, such as at a depth of 3 m, the foundation pit will experience toe—kick phenomenon.

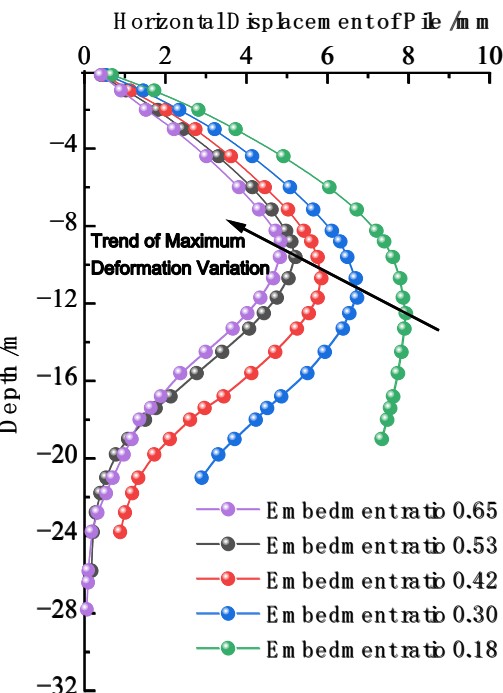

**Figure 15.** Horizontal displacement of retaining pile with different embeddedness ratio.

To further quantify the influence of different embedment ratios on the maximum lateral displacement of the retaining piles, extract the maximum lateral displacement value of the retaining piles under different embedment ratios and obtain the influence of different embedment ratios on the maximum lateral displacement of the retaining piles as shown in Figure 16. Within a certain range, as the embedment ratio increases, the rate of decrease of the maximum lateral displacement of the retaining piles is approximately an exponential function. After the embedment ratio reaches a certain value, it no longer has an effect on the deformation of the retaining piles. A smaller embedment ratio is more likely to cause the "toe-kick" risk of the foundation pit. Based on the actual engineering geological conditions, the depth of the retaining pile into weathered rock should be 9 m. From the perspective of excavation safety, the embedment ratio should not be lower than about 0.53, to avoid causing instability and damage to the excavation pit.

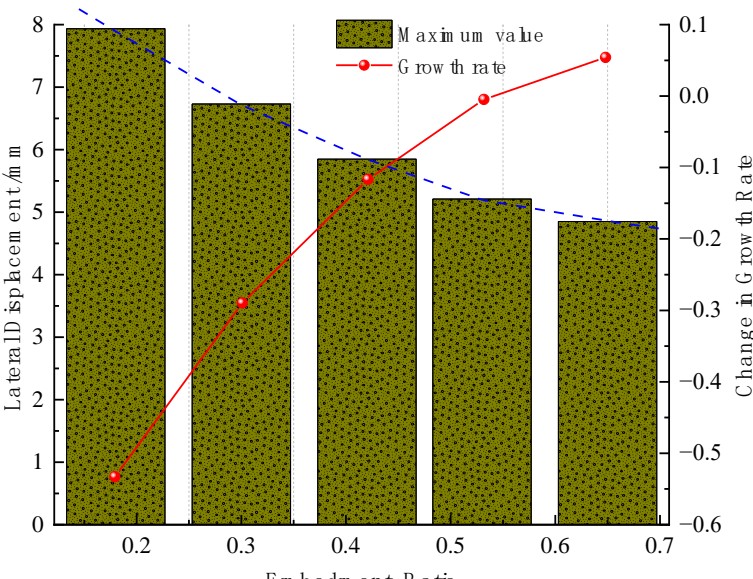

**Figure 16.** Analysis of maximum deformation of retaining pile with different embedment ratio.

### 6.2. *Analysis of the Influence of Geological Parameters*

6.2.1. Analysis of the Influence of Internal Friction Angle

To analyze the deformation response characteristics of the retaining pile under the coupled effect of foundation pit dewatering and different geological parameters on the influence area of the fault zone, the method of controlling variables was adopted to keep the other variables constant and analyze the influence of one variable. Based on the engineering design parameters in this paper, while keeping the other geological parameters constant, the internal friction angle was adjusted to $0.6\varphi$, $0.8\varphi$, $\varphi$ and $1.2\varphi$ in turn, and the influence of the change in internal friction angle on the deformation of the retaining pile was studied. As shown in Figure 17, the deformation law curve of the retaining pile with the change of internal friction angle was obtained. Under different internal friction angles, the deformation of the retaining pile excavated by the foundation pit dewatering showed a "bulging" curve law. With the increase of internal friction angle, the frictional resistance between soil particles increased, the displacement of the soil on the pile side decreased, resulting in the contraction of the retaining pile deformation, and within a certain range, the rate of deformation reduction decreased, and the deformation amount was negatively correlated with the increase of the internal friction angle increment.

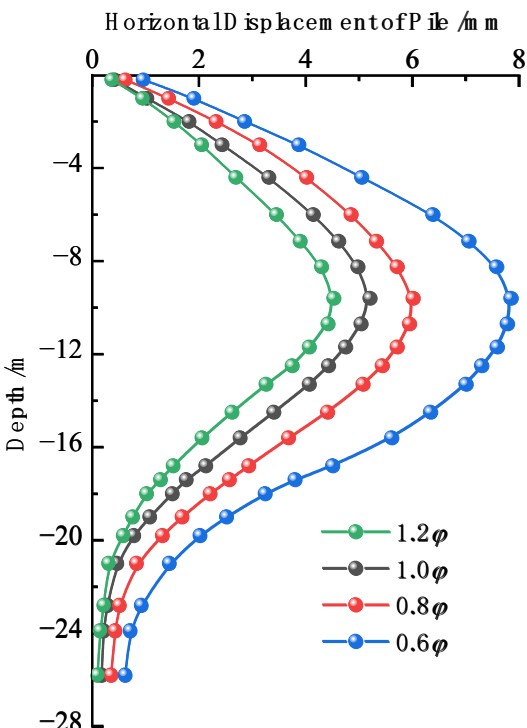

**Figure 17.** Horizontal displacement of retaining pile with different frictional angle.

6.2.2. Analysis of the Influence of Cohesion

While keeping the other geological parameters constant, the cohesion was adjusted to $0.6\ C$, $0.8\ C$, $C$ and $1.2\ C$ in turn, and the influence of the change in cohesion on the deformation of the retaining pile was studied. As shown in Figure 18, the deformation law curve of the retaining pile with the change of cohesion was obtained. Under different cohesion, the deformation of the retaining pile excavated by the foundation pit dewatering showed a "bulging" curve law. With the increase of cohesion, the physical and chemical interactions between soil particles increased, the displacement of the soil in the active zone on the pile side decreased, resulting in the contraction of the retaining pile deformation, and within a certain range, the rate of deformation reduction decreased, and the deformation amount was negatively correlated with the increase of the cohesion increment.

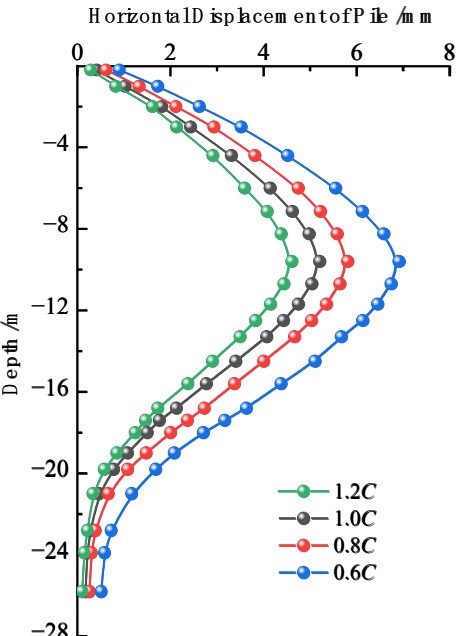

**Figure 18.** Horizontal displacement of retaining pile with different Cohesion.

6.2.3. Analysis of the Influence of Elastic Modulus

While keeping the other geological parameters constant, the elastic modulus was adjusted to 0.6 *E*, 0.8 *E*, *E* and 1.2 *E* in turn, and the influence of the change in elastic modulus on the deformation of the retaining pile was studied. As shown in Figure 19, the deformation law curve of the retaining pile with the change of elastic modulus was obtained. Under different elastic modulus, the deformation of the retaining pile excavated by the foundation pit dewatering showed a "bulging" curve law. With the increase of elastic modulus, the displacement of the soil in the active zone on the pile side decreased, resulting in the contraction of the retaining pile deformation, and within a certain range, the rate of deformation reduction decreased, and the deformation amount was negatively correlated with the increase of the elastic modulus increment.

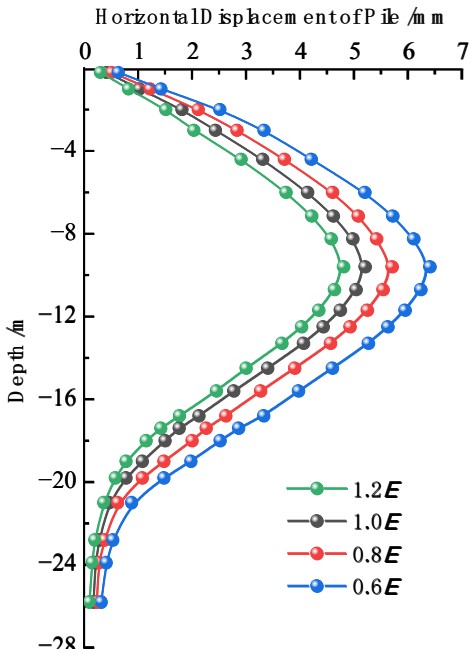

**Figure 19.** Horizontal displacement of retaining pile with different elasticity modulus.

6.2.4. Sensitivity Analysis

Let a system $\gamma$ be determined by n main influencing parameters ($\alpha_1$, $\alpha_2$, $\alpha_3$, ... , $\alpha_n$). The relationship expression between variable $\gamma$ and parameter $\alpha$ is $\gamma = f$ ($\alpha_1$, $\alpha_2$, $\alpha_3$, ... , $\alpha_n$). Assuming that under a certain condition, the parameters are set to baseline parameters, the system baseline variable $\gamma'$ is obtained as $f$ ($\alpha_1'$, $\alpha_2'$, $\alpha_3'$, ... , $\alpha_n'$). By adjusting the changes in different influencing parameters on the system variable, the trend and degree of deviation of the system variable $\gamma$ from the baseline variable $\gamma'$ are analyzed, which is called sensitivity analysis [30] of the parameters.

To facilitate sensitivity analysis among different parameters, dimensionless sensitivity functions and sensitivity factors are defined. Specifically, $\gamma$ and $\alpha_i$ are defined as the ratio of the relative error, denoted as the sensitivity function $S_i(\alpha_i)$, with an approximate expression as shown in Equation (11):

$$S_i(\alpha_i) \left| \frac{df(\alpha_i)}{d(\alpha_i)} \right| \frac{\alpha_i}{\gamma} (k = 1, 2, 3 \dots) \tag{11}$$

By substituting $\alpha_i'$ into Equation (11), the sensitivity factor $S_n$ for $\alpha_i'$ is obtained. Under the baseline state, the sensitivity of $\gamma$ to $\alpha i$ increases with the increase of $S_n$. In comparison with the size of the sensitivity factor $S_n$, the sensitivity is stronger when the sensitivity factor is larger and weaker when it is smaller. This paper conducts sensitivity analysis on geological parameters, namely, internal friction angle $\varphi$, cohesion $C$, and elastic modulus $E$.

To determine the strength of the influence of geological parameters, including internal friction angle $\varphi$, cohesion $C$ and elastic modulus $E$, on the maximum deformation of retaining piles under the coupling effect of excavation and dewatering in the influence area of the fault zone, sensitivity analysis of the maximum displacement of retaining piles under different geological parameters is performed.

The sensitivity curves of different geological parameter changes on the maximum deformation of retaining piles are shown in Figure 20. Under different geological parameters, the decreasing trend of the maximum deformation of retaining piles is inconsistent. With the increase of geological parameters, the maximum deformation of retaining piles shows a non-linear decrease, and within a certain range, the rate of deformation contraction shows a decreasing trend. Among them, the influence curve of internal friction angle on the deformation of retaining piles during excavation is the most prominent, followed by cohesion and elastic modulus. By fitting the sensitivity curves of different parameter changes on the maximum deformation of retaining piles with a quadratic function, the sensitivity function $S(x)$ of the maximum deformation of retaining piles under different geological parameters is obtained. By calculating the formula, the sensitivity factors of different geological parameters on the maximum deformation of retaining piles are obtained and shown in Table 4. The sensitivity factor of the internal friction angle is obviously greater than that of cohesion and elastic modulus, indicating that the sensitivity of the internal friction angle to the maximum deformation of retaining piles during excavation is higher. The most sensitive factor affecting the maximum deformation of retaining piles during excavation is the internal friction angle, followed by cohesion and elastic modulus.

**Table 4.** Sensitive factor summary.

| Indicators | Sensitive Factor $S_n$ |
|---|---|
| Internal Friction Angle $\varphi$ | 0.760 |
| Cohesion $C$ | 0.604 |
| Elasticity Modulus $E$ | 0.422 |

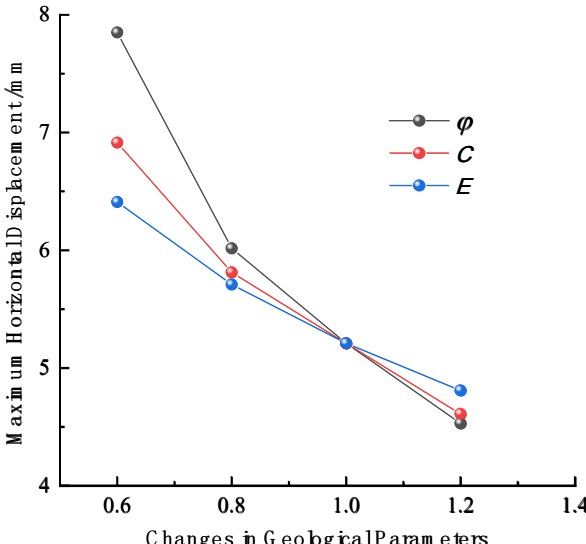

**Figure 20.** Comparison of sensitivity of geological parameters to maximum deformation of retaining pile.

## 7. Structural Design Optimization

The original design of the excavation pit's retaining piles had small deformations but was too conservative in its design. Therefore, from the perspectives of safety and cost-effectiveness, it is necessary to optimize the excavation pit support design. Considering the good geological conditions in the area and the high stiffness of the support system, the optimization was carried out by changing the concrete struts in the original design to steel struts and adjusting the diameter of the retaining pile from 1.2 m to 1.0 m. The deformation curve of the optimized retaining pile is shown in Figure 21. After optimization, the lateral displacement curve of the retaining pile remains basically the same, but the maximum deformation position has shifted slightly downwards due to the decrease in the stiffness of the support system. The maximum deformation of the retaining pile has increased from 5.21 mm to 13.63 mm, which obviously meets the control requirements of the specification (0.3%*H*), and is controlled within a lower deformation range. This optimization achieves the purpose of reducing investment and shortening the construction period and has certain practical engineering significance.

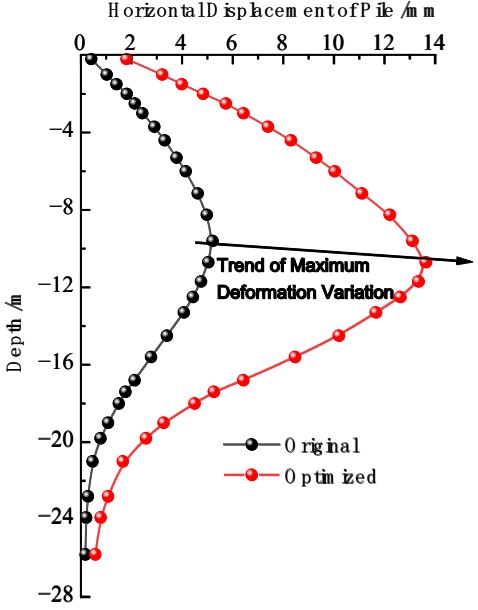

**Figure 21.** Comparison of horizontal displacement of retaining pile before and after optimization.

## 8. Conclusions

This paper analyzes the deep excavation pit project in Luohu, an area in Shenzhen that is affected by the typical F1322 fault zone, as an example, and conducts research on the deformation characteristics of the retaining pile under the effects of dewatering excavation through theoretical calculations, numerical simulations and on-site measurements. The following conclusions were mainly drawn:

(1) Based on the small deformation theory, it is proposed to consider the coupled effect of excavation and dewatering during foundation pit construction and use the energy method of elasticity theory to solve the analytical solution for the deformation of retaining piles, which yields more desirable calculation results.

(2) By comparing and analyzing the results of the retaining wall deformation theory calculation, finite element calculation, and field measurement data, the numerical rules are basically consistent. Simplified calculation only considers rotational deformation and ignores the translational deformation of the wall, resulting in a large deviation between the theoretical calculation results of the wall bottom deformation and the measured values. To reduce the deviation between numerical results and measured values, strict measures such as "zone excavation, avoiding peripheral loading, and optimizing construction deployment" should be taken in pit construction, strengthen construction monitoring, and reduce the impact on retaining wall deformation.

(3) The maximum deformation growth rate $k$ ($\Delta S_{max}/\Delta$) of the retaining wall decreases exponentially with the increase of the structural stiffness parameters ($E$ and $I$) and the embedment ratio in a certain range. To ensure the safety of retaining wall deformation, the retaining wall design must have a certain reserve of stiffness and embedment ratio.

(4) With the increase of geological parameters, the lateral displacement of the retaining wall gradually decreases, and the decreasing rate gradually decreases within a certain range, that is, the maximum deformation of the retaining wall has a nonlinear relationship with the geological parameters. A quadratic function is used to fit the sensitivity function $S(x)$ of the maximum lateral displacement of the retaining structure to the changes in various geological parameters, and the sensitivity analysis of geological parameters is carried out. It is found that the internal friction angle is the most sensitive factor, followed by cohesion, and the elastic modulus is the smallest.

(5) The structural optimization plan includes replacing the original concrete struts with steel struts, adjusting the diameter of the perimeter piles from 1.2 m to 1.0 m, and increasing the maximum deformation of the perimeter piles from 5.21 mm to 13.63 mm to meet the specification's ($0.3\%H$) limit. The optimization of the excavation support system not only enables compliance with the standard construction procedures but also reduces investment and shortens construction time, fully aligning with the current design principles of safety, economy, and sustainable development.

**Author Contributions:** Conceptualization, Y.N.; Software, Y.N.; Formal analysis, Y.N.; Investigation, Q.W.; Writing—original draft, Y.N.; Writing—review & editing, F.M.; Project administration, L.Z.; Funding acquisition, F.M. All authors have read and agreed to the published version of the manuscript.

**Funding:** This research was funded by the National Natural Science Foundation of China (51474045).

**Data Availability Statement:** Not applicable.

**Conflicts of Interest:** The authors declare no conflict of interest.

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
