# Peer review of "Study on Deformation Characteristics of Retaining Structures under Coupled Effects of Deep Excavation and Groundwater Lowering in the Affected Area of Fault Zones"

_sustainability, doi:10.3390/su15108060_

Round 1
Reviewer 1 Report
I have only minor comments on this manuscript.
Lines 48-51 - this is a very good place for some proper references.
Lines 112-114 - exactly what kind of fault this one is - low-angle/high-angle, normal/reverse, strike-slip (dextral/sinistral), oblique, etc. This is very important for structural engineering as specific type of fault and motion character along fault plane will severely influence rock mechanics and properties of surrounding soils.
Line 116 - what exactly (in this case) is "plain fill soil". Is this just fill dirt used to patch holes in the ground or create positive topography in landscaping? What is its relationship with top soil, if any? Please comment.
Line 117-119 - any organic material within this soil profile? Any evidence for presence of organic horizon?
Angular gravel - size range? Please comment.
Line 118 - highly weathered rock - what kind of weathering? Any iron minerals present - Fe-oxides or hydroxides present (limonite, hematite, etc.), which are sensitive to groundwater movement/withdrawal and can cause dangerous slippages or other forms of mass wasting.
Author Response
Dear Dr,
Thank you very much for taking the time to review our manuscript entitled "Study on Deformation Characteristics of Retaining Structures under Coupled Effects of Deep Excavation and Groundwater Lowering in the Affected Area of Fault Zones." We appreciate your valuable comments and suggestions, which have helped us to improve the quality of our work.
We have carefully considered all of your comments and suggestions, and have made the following revisions to the manuscript:
- Relevant literature has been added.
- F1322 is a normal fault.
- Regarding the translation issue,"plain fill soil" should be changed to "uncompacted fill soil". In reality, it refers to the stratification of the entire site from top to bottom.
- There is no organic matter soil in the stratigraphic profile, and the diameter of angular gravel ranges from 2mm to 8mm.
- The gravelly clayey soil is completely derived from the weathered residual rock beneath it, and there are no iron minerals present in the weathered material.
We hope that these revisions address your concerns and have strengthened the manuscript. We have also attached a marked-up version of the revised manuscript to this email to show you exactly what changes have been made.
Once again, we appreciate your insightful feedback and the time you have dedicated to reviewing our work. Please let us know if you have any further questions or concerns.
Sincerely,
Yungang Niu
Reviewer 2 Report
Dear Authors,
I would recommend this manuscript after addressing the minor revisions. Please find my comments in detail attached.

The manuscript needs minor English editing.
Author Response
Dear Dr,
Thank you very much for taking the time to review our manuscript entitled "Study on Deformation Characteristics of Retaining Structures under Coupled Effects of Deep Excavation and Groundwater Lowering in the Affected Area of Fault Zones." We appreciate your valuable comments and suggestions, which have helped us to improve the quality of our work.
We have carefully considered all of your comments and suggestions, and have made the following revisions to the manuscript:
- Already supplemented.
- Already modified and supplemented.
- Already supplemented.
- Liu, G.B.; Wang, W.D. Handbook of Foundation Pit Engineering (2nd edition). M. Beijing: China Architecture & Building Press, 2009: 2-3.
- Already supplemented.
- Yes, it has been modified in the article.
- Already modified.
- Already supplemented.
- Li, E.B.; Tan, Y.H.; Zhang, S.G.; Ding, L.L. Analytical calculation of deformation of diaphragm wall in Deep foundation pit Enclosure. J. Journal of PLA University of Science and Technology, 2004(02):57-60.
During the plastic deformation stage of the soil, due to the rearrangement of the particles inside the soil and the adjustment of the stress state, the physical properties such as volume, shape, and mass of the soil may change over time. This time-dependent plastic deformation characteristic is called time effect. Therefore, when the soil is in a plastic state, due to the influence of time effect, the measured deformation increases, but the rate of deformation growth gradually slows down with time, causing the measured value to be larger than the calculated value.
- Zhang, A.Q.; Wu, A.X.; Han Bin, et al. A new model for the thickness of filled retaining wall based on the theory of elastic thin plates. J. Journal of Central South University (Science and Technology), 2018, 49(03): 696-702.
- Ciarlet, P.G.; Mardare, C. The intrinsic theory of linearly elastic plates. Mathematics and Mechanics of Solids, 23(2), 2018.
- Gu, W.; Zhang, L.Y.; Tan, Z.X.; Deng, K.Z. Stability analysis of open stope roof based on the model of elastic thin plates. J. Journal of Mining and Safety Engineering, 2013, 30(06): 886-891.
- Already supplemented.
- Zheng, J.H. Stiffness analysis of circular section concrete retaining piles. J. Fujian Architecture, 2005(Z1):232-233+221.
We hope that these revisions address your concerns and have strengthened the manuscript. We have also attached a marked-up version of the revised manuscript to this email to show you exactly what changes have been made.
Once again, we appreciate your insightful feedback and the time you have dedicated to reviewing our work. Please let us know if you have any further questions or concerns.
Sincerely,
Yungang Niu
Reviewer 3 Report
The manuscript has a detailed investigation about the Deformation Characteristics of Retaining Structures under Coupled Effects of Deep Excavation and Groundwater Lowering in the Affected Area of Fault Zones. A very interesting work. Here is some comments
1 Please add the following sentence in Line 95
"Savvides and Papadopoulos formed a Feed Forward Neural Network that estimate failure stresses and strains in Shallow Foundations formed through Stochastic Finite Element Analysis following Savvides and Papadrakakis"
Where the references are
Savvides, A.-A.; Papadopoulos, L. A Neural Network Model for Estimation of Failure Stresses and Strains in Cohesive Soils. Geotechnics 2022, 2, 1084-1108. https://doi.org/10.3390/geotechnics2040051
Savvides A A Papadrakakis M. A computational study on the uncertainty quantification of failure of clays with a modified Cam-Clay yield criterion. SN Appl. Sci. 3, 659 (2021). https://doi.org/10.1007/s42452-021-04631-3
2 Please Correct Figure 15 X Axis. Has a misspelling of Embedding.
3 Please enter a small explanation-definition for embedding ratio. It is easier for the reader.
4 Please enter the initial values for c, φ, E and provide appropriate explanations for the selection of the values.
5 In the analysis of structural optimization how the objective function and the constraints are defined? What are the economic or other factors that influence the largest accepted displacement and deformation? Please provide a brief but clear presentation of the parameters set to the optimization procedure and method of solution.
Minor corrections should be done
Author Response
Dear Dr,
Thank you very much for taking the time to review our manuscript entitled "Study on Deformation Characteristics of Retaining Structures under Coupled Effects of Deep Excavation and Groundwater Lowering in the Affected Area of Fault Zones." We appreciate your valuable comments and suggestions, which have helped us to improve the quality of our work.
We have carefully considered all of your comments and suggestions, and have made the following revisions to the manuscript:
- Related content and references have been added.
- It has been modified.
- Embedment ratio = Depth of pile embedment / Depth of excavation of foundation pit. And explained in the text
- The initial values of C, φ, and E are detailed in Table 1 of the main text, and these parameters are obtained from the geotechnical test report.
- To evaluate the indicators of the functionality of the foundation pit support structure, the objective function is defined as:
V(x) = F(x) / C(x)
Where F(x) is the functionality evaluation function, C(x) is the cost evaluation function.
The variables with the greatest impact on the transformation are: internal support strength, retaining pile diameter, and embedment ratio.
Constraints: According to design specifications and actual engineering conditions, the maximum deformation of the retaining pile cannot exceed 0.3%H.
This article aims to optimize the structural transformation by reducing costs, adjusting structural functionality, and improving the value coefficient of the structure.
We hope that these revisions address your concerns and have strengthened the manuscript. We have also attached a marked-up version of the revised manuscript to this email to show you exactly what changes have been made.
Once again, we appreciate your insightful feedback and the time you have dedicated to reviewing our work. Please let us know if you have any further questions or concerns.
Sincerely,
Yungang Niu